

# Alignment of scanning lidars in offshore wind farms

Andreas Rott[1], Jörge Schneemann[1], Frauke Theuer[1], Juan José Trujillo Quintero[2], and Martin Kühn[1]

[1]ForWind, Institute of Physics, Carl von Ossietzky University Oldenburg, Küpkersweg 70, 26129 Oldenburg, Germany
[2]UL International GmbH, Kasinoplatz 3, 26122 Oldenburg, Germany

**Correspondence:** Andreas Rott (andreas.rott@uni-oldenburg.de)

**Abstract.** Long-range Doppler wind lidars are applied more and more for high resolution areal measurements in and around wind farms. Proper alignment, or at least knowledge on how the systems are aligned, is of great relevance here. The paper describes in detail two methods that allow a very accurate alignment of a long-range scanning lidar without the use of extra equipment or sensors. The well-known so-called *Hard Targeting* allows a very precise positioning and north alignment of the lidar using the known positions of the surrounding obstacles, e.g. wind turbine towers. Considering multiple hard targets instead of only one with a given position in an optimization algorithm allows to increase the position information of the lidar device and minimizes the consequences of using erroneous input data. The method, referred to as *Sea Surface Leveling*, determines the leveling of the device during offshore campaigns in terms of roll and pitch angle based on distance measurements to the water surface. This is particularly well suited during the installation of the systems to minimize alignment error from the start, but it can also be used remotely during the measurement campaign for verification purposes. We applied and validated these methods to data of an offshore measurement campaign, where a commercial long-range scanning lidar was installed on the transition piece platform of a wind turbine. In addition, we present a model that estimates the quasi-static inclination of the device due to the thrust loading of the wind turbine at different operating conditions. The results show reliable outcomes with a very high accuracy in the range of $0.02°$ in determining the leveling. The importance of the exact alignment as well as the possible applications are discussed in this paper. In conclusion, these methods are useful tools that can be applied without extra effort and contribute significantly to the quality of successful measurement campaigns.

## 1 Introduction

Scanning long-range Doppler wind lidar devices play an increasingly important role in the assessment of wind conditions (Krishnamurthy et al., 2016). Due to their ability to measure the wind speed over long distances and over an entire area with high temporal and spatial resolution, it is possible to obtain knowledge about wind fields. This can be utilized for wind resource assessment offshore from the coast (Koch et al., 2012; Shimada et al., 2020) or existing offshore platforms and wind energy research on e.g. wind turbine wakes (Krishnamurthy et al., 2017), wind farm cluster wakes (Schneemann et al., 2020a), meteorological phenomena like low level jets (Pichugina et al., 2016) or global wind farm blockage (Schneemann et al., 2020b) and minute-scale wind power forecasts to improve grid stability and energy trading (Theuer et al., 2020).

The wind field in the lowest part of the (marine) boundary layer is by nature inhomogeneous. Flow complexity is exacerbated around and inside wind farms. For wind energy applications the characterisation of the wind conditions has to be performed



with a spatial accuracy of some meters. Therefore, an exact positioning and orientation of the measuring device during a measurement campaign is very important. This is a major challenge for remote sensing devices, such as lidars, since even small inclination errors can lead to large deviations, e.g. in the resulting measurement height. To give an example, an error of 30    $0.25°$ in alignment results in an altitude error of about $44\,\mathrm{m}$ at $10\,\mathrm{km}$ distance.

The position of a lidar device can be assessed with an internal GPS or by surveying methods with sufficient accuracy. However, the orientation of the system, which is more critical, is typically determined with two sensors with unknown (arguably insufficient) accuracy. The full orientation in three-dimensional space is given by three rotation angles, namely, bearing, pitch and roll. The bearing represents the deviation from North on the horizontal plane and is measured by a compass, as synonyms for this we also use the terms "northing" and "orientation" in this paper. Pitch and roll, which mainly affect the elevation of the laser beam, are measured by an internal inclinometer or a level spirit. To our knowledge, lidar manufacturers do not supply any calibration protocol of such sensors. Therefore, in a campaign they can only be used for rough leveling, and must be complemented by more accurate measurements.

Doppler wind lidars are able to detect even very little backscatter from aerosols. Therefore, measurements against hard targets result in very high Carrier-to-Noise Ratios (CNR) in the detected signal. This phenomenon can be utilized to determine the distance of a hard target from the lidar and its direction with respect to the scanner orientation of the lidar, i.e. its line-of-sight. An established technique to determine the positioning and the orientation of the device is the so-called *Hard Targeting* (HT). This technique can be applied onshore as well as offshore. It uses existing hard targets with known positions like met masts, light houses or wind turbine towers for north orientation. Vasiljevic et al. (2014) introduced a method called *CNR mapper* allowing to check the north orientation and the tilt of the lidar by mapping the CNR around hard targets with well known position and height using several RHI or PPI scans.

In an offshore wind farm setting it is difficult to get hard targets with a fixed height. From a lidar perspective, wind turbines change in height due to the two main moving components, nacelle and rotor blades. This makes scanning setup and data analysis very difficult for the estimation of the leveling of the device. To overcome this, Rott et al. (2017) introduced a method called *Sea Surface Leveling* (SSL) for offshore installed lidars. This method does not rely on external object heights but uses the sea surface as a reference. This method reaches a much higher accuracy with several technical advantages. The scanning setup is very simple with almost no need for setup. Additionally, the reference is ubiquitous and no pre-processing of external and uncertain information is needed. This allows to perform leveling on the fly at installation time. In fact, it has been applied to effectively minimize installation errors (Trujillo et al., 2019). This is of practical advantage in offshore campaigns, where installation time windows are very tight. Furthermore, SSL can be applied during commissioning for calibration of any internal leveling sensor (Trujillo et al., 2019, 2021) and accurately correct scanning trajectories. Finally, the procedures are fast and can be applied continuously in the measurement program without causing long interruptions of the main scanning strategy. This can be applied for orientation monitoring purposes during the whole measurement campaign.

For HT no commonly used method has been established so far; in most cases researchers use individual estimations. The *CNR mapper* is not fully compatible with all lidar softwares. Furthermore, at offshore sites often only wind turbines are available as hard targets not offering a defined hard target in a well known height due to rotor and yaw movements.





Knowing the leveling of the device becomes particularly relevant when a lidar is installed on a moving base, as discussed previously in (Gottschall et al., 2014). Bromm et al. (2018) explained that measurements from the top of the nacelle of a wind turbine prove difficult because the nacelle tilts due to thrust. This is because the lidar support platform changes its tilt
dynamically depending on atmospheric and operational conditions. This situation is also given, although less pronounced, in an installation at the transition piece of a turbine. Due to the high accuracy needed in leveling, this also needs to be quantified. Two approaches are conceivable for this. First, if the lidar system has an inclinometer, it can be calibrated with SSL and then used. Second, if a reliable inclinometer is not available, an empirical model of the platform's inclination can be adapted. An approach to derive such a model is presented here. The *Platform Tilt Model* (PTM) was developed based on standard signals
from the turbine SCADA system.

With the three concepts introduced, a framework for accurately assessing the northing and leveling of scanning lidar equipment installed on elevated offshore structures can be achieved. Therefore, the goal of our work is to present the concepts, their implementation and validation in a real offshore wind farm environment. The paper is structured following the three objectives:

1. Determination of the accurate northing and positioning of a lidar based on measurements with the device. This is achieved
75       with the help of *Hard Targeting* (HT). This method is not limited to being used only offshore.

2. Precise measurement of the inclination of an offshore installed lidar using *Sea Surface Leveling* (SSL). Explanation of this method and extension so that device-specific parameters can be better considered.

3. Estimation of the live roll and pitch angles of a lidar installed on the transition piece of a wind turbine based on operational measurements of the wind turbine using a *Platform Tilt Model* (PTM).

**2   Methods**

The three methods of HT, SSL and the PTM involve several steps, outlined in Figure 1. They are briefly summarized before going into more detail in the following sections. The grey boxes represent the HT, which is described in more detail in Section 2.1. Starting from the top, the *Hard Target Scans* are performed in the first step. Based on these scans we use the HT to acquire the northing $\gamma_0$ and position $x_0, y_0$ of the lidar. The blue boxes demonstrate the procedure of the SSL, which is
detailed in Section 2.2. The first step is to perform the sea surface level scans. Based on them, we use the *Sea Surface Distance Estimation* as described in Section 2.2.1, to determine the *Sea Surface Intersection* $r_{\mathrm{sea}}$. The SSL uses the *Sea Surface Model*, shown in Section 2.2.2, to estimate the respective roll and pitch angles $\rho_{\mathrm{m}}, \phi_{\mathrm{m}}$ by following Section 2.2.3. Next, we apply the PTM, illustrated by the green boxes, introduced in Section 2.3, to these results in combination with the available SCADA data to approximate the resting roll and pitch angles $\rho_{\mathrm{r}}, \phi_{\mathrm{r}}$ as well as a linear scaling parameter $c$. Finally, we can use the *Inverse*
*Platform Tilt Model* to predict the leveling of the lidar in terms of live roll and pitch $\rho_{\mathrm{WT}}, \phi_{\mathrm{WT}}$, from the SCADA Data. To illustrate the methods, we already use sample data from the measurement campaign described in Section 2.4.



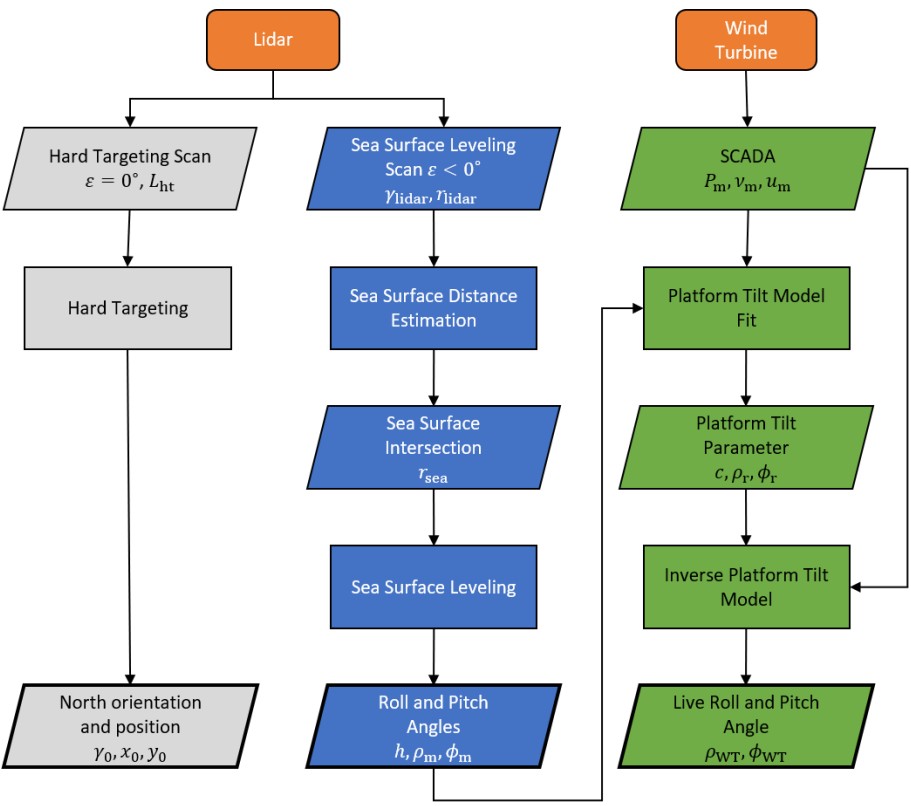

**Figure 1.** Overview of the procedure of the methods. In orange are the input devices. HT is shown in grey, SSL in blue and PTM in green. The variables are explained in the following Sections.

## 2.1 Hard Targeting (HT)

HT is a method to determine an accurate orientation and positioning of the lidar based on fixed objects in the vicinity whose positions are known. The method is outlined by the grey boxes in Figure 1. The lidar measures information about the quality

of the signal in form of the CNR $\xi_{\mathrm{cnr}}(\gamma_{\mathrm{lidar}}, r_{\mathrm{lidar}})$ in each measurement location with the azimuth angle $\gamma_{\mathrm{lidar}} \in [0°, 360°[$ and the range $r_{\mathrm{lidar}} \in \mathbb{R}_{\geq 0}$. Solid objects yield stronger backscatter than the aerosols in the air leading to a much higher CNR value. This makes it possible to identify the location of these obstacles in the measurements. For the HT we performed plan position indicator (PPI) scans with an elevation of $\varepsilon_{\mathrm{ht}} = 0°$, i.e. horizontal PPI scans. First, we used a relatively fast 360° PPI scan to identify the rough location of viable hard targets, followed by higher resolution scans targeted at those locations. For

these latter scans we set the angular resolution of the azimuth angle to $\Delta\gamma_{\mathrm{lidar}} = 0.1°$ and the spatial resolution of the range gates to $\Delta r_{\mathrm{lidar}} = 2\,\mathrm{m}$.

Figure 2 shows the CNR value of a HT-Scan for a single beam , where a solid object was hit by the laser beam, in comparison to the median CNR curve for all azimuth angles of the respective scan. The peak in the signal represents the quasi Gaussian shape of the probe length volume, i.e. the laser pulse. The exact object range depends on the shape and symmetry of the pulse



of the lidar system. However, a good approximation is the center. It is to note, that the method to estimate the northing shown in the next section is resilient to inaccuracies in this distance.

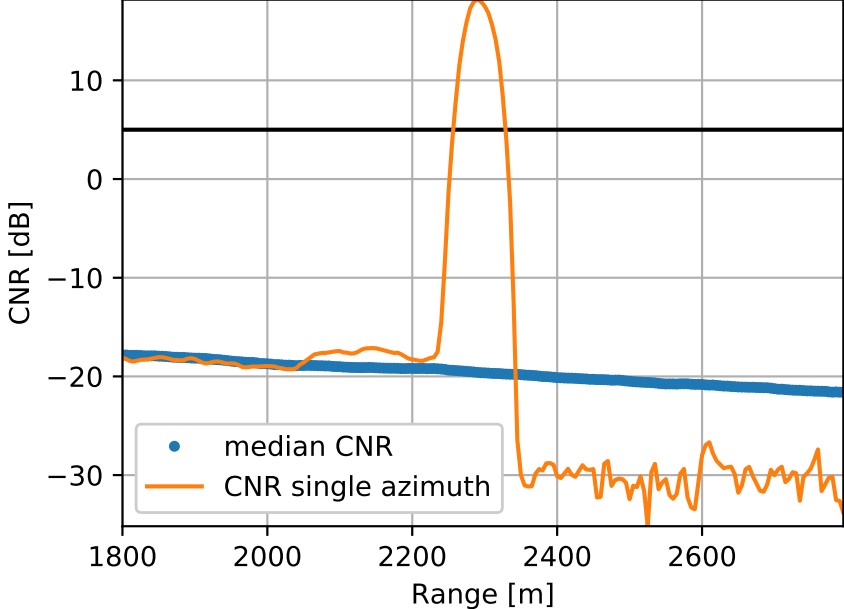

**Figure 2.** Example of the measured CNR over the range for a hard target scan for a single beam (orange) compared to the median for all azimuth angles (blue). The black horizontal line presents the $5\,\mathrm{dB}$ threshold.

In a wind farm, it makes sense to use the towers of the other turbines as targets for the hard target scans, as we did in the case presented here. The turbine locations are given in a Cartesian coordinate system and are denoted by $(x_{\mathrm{wt},i}, y_{\mathrm{wt},i}) \in \mathbb{R}^2$ for $i = 1, \ldots, N_{\mathrm{wt}}$ with $N_{\mathrm{wt}} \in \mathbb{N}$ being the number of turbines. The first step is to identify the targets in the measurements. We use

a CNR filter for this. For the WindCube 200S we chose a simple threshold for the CNR $\xi_{\mathrm{cnr}}(\gamma_{\mathrm{lidar}}, r_{\mathrm{lidar}})$. We define the set of locations $L_{\mathrm{ht}}$ of the hard target measurements as positions in the polar coordinate system of the lidar where $\xi_{\mathrm{cnr}}(\gamma_{\mathrm{lidar}}, r_{\mathrm{lidar}}) \in \mathbb{R}$ is greater or equal to $5\,\mathrm{dB}$.

$$L_{\mathrm{ht}} := \{ (\gamma_{\mathrm{lidar}}, r_{\mathrm{lidar}}) \mid \xi_{\mathrm{cnr}}(\gamma_{\mathrm{lidar}}, r_{\mathrm{lidar}}) \geq 5\,\mathrm{dB} \}. \tag{1}$$

If the resolution of the scan is fine enough, a solid object, such as the tower of a wind turbine, is represented by a cluster of

hard target measurements. Let $N_{\mathrm{ht}} := \#L_{\mathrm{ht}}$ be the cardinality of the identified hard target measurements $L_{\mathrm{ht}}$.

In the next step, we are constructing an optimization, which minimizes the distances of the identified hard targets to the known locations by adjusting the three parameters $x_0, y_0$ and $\gamma_0$, which represent the position and the north orientation of the lidar, respectively. Hard targets that were identified in the CNR values, but for which no coordinates are known should be excluded from the process, because the optimization tries to minimize the distances to the known locations.

The cost function is defined as:

$$
C_{\mathrm{ht}}(x_0, y_0, \gamma_0) \;=\; \sum_{j=1}^{N_{\mathrm{ht}}} \left( \min_{i=1,\dots,N_{\mathrm{wt}}} \left( (x_{\mathrm{wt},i} - x_{\mathrm{ht},j}(x_0,\gamma_0))^2 + (y_{\mathrm{wt},i} - y_{\mathrm{ht},j}(y_0,\gamma_0))^2 \right) \right) \tag{2}
$$

$$
\text{with}:\quad x_{\mathrm{ht},j}(x_0,\gamma_0) \;=\; \sin(\gamma_{\mathrm{ht},j} + \gamma_0) \cdot r_{\mathrm{ht},j} + x_0 \tag{3}
$$

$$
y_{\mathrm{ht},j}(y_0,\gamma_0) \;=\; \cos(\gamma_{\mathrm{ht},j} + \gamma_0) \cdot r_{\mathrm{ht},j} + y_0 \tag{4}
$$

$$
(\gamma_{\mathrm{ht},j}, r_{\mathrm{ht},j}) \;\in\; L_{\mathrm{ht}} \tag{5}
$$

The cost function calculates the quadratic Euclidean distances of the identified hard targets to all known locations of the turbines, chooses the turbine closest to the respective hard target and sums up all the squares of the closest distances. We can now formulate the optimization problem as

$$
\min_{x_0,y_0,\gamma_0} \quad C_{\mathrm{ht}}(x_0, y_0, \gamma_0) \tag{6}
$$

$$
\text{s.t.}:\quad x_0, y_0 \in \mathbb{R}, \quad \gamma_0 \in [0°, 360°[ \tag{7}
$$

This optimization can be solved numerically quite easily. We used the Nelder-Mead algorithm (Gao and Han, 2012) for nonlinear optimization problems provided by the minimize function of Python's SciPy module (Virtanen et al., 2020). This algorithm is a downhill simplex method that starts with an initial guess for the solution, which must be chosen by the user. This solution should ideally be quite close to the optimal solution in order for the algorithm to converge to the correct minimum.

## 2.2    Sea Surface Leveling (SSL)

SSL is a method that uses the water surface to estimate the leveling of the lidar. An overview is given in Figure 1 in the blue boxes.

### 2.2.1    Sea Surface Leveling Scans

For the SSL we measure the distance from the scanner head to the sea surface for different azimuth angles by the use of a PPI scan with a negative elevation. In our case we set the elevation to $\varepsilon_{\mathrm{ssl}} = -3°$. We had a sector of approximately $268°$ with
line of sight from the lidars to the sea surface, while the rest was blocked by the transition piece and the turbine tower. We set the range gates of the device to $r_{\mathrm{lidar}} \in \{400\ \mathrm{m}, 401\ \mathrm{m}, \dots, 549\ \mathrm{m}\}$ in accordance with the roughly estimated distance of the scanner to the sea surface. The *Sea Surface Leveling Scans* were carried out on 11 and 12 April 2019, 2 and 3 May 2019 and from 14 to 17 May 2019. Figure 3 illustrates the *Sea Surface Leveling Scan*.

We assume that the sea surface is a flat and horizontal area, neglecting the curvature of the earth, and we interpret waves as
measurement noise. The shape of the intersection between this scan and the sea surface is a conic section with an azimuthal range of less than $360°$ due to the tower shadow and provides us with information about the leveling of the lidar. For example, if the leveling of the device is accurate the intersection results in a nearly perfect circle, with relatively small fluctuations due to waves. If the leveling is slightly misaligned the shape of the intersections becomes an ellipse. For larger misalignment the



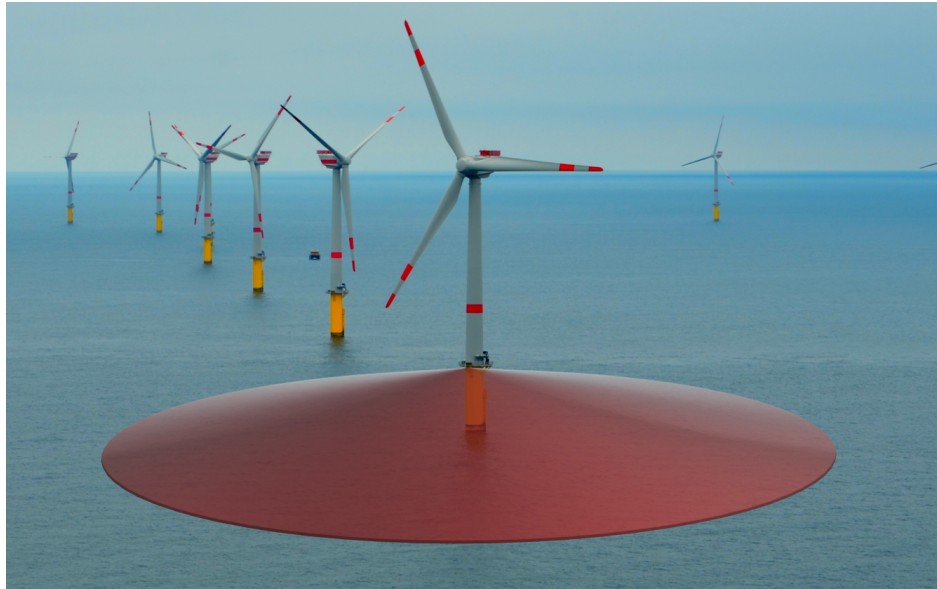

**Figure 3.** Schematic representation of the *Sea Surfac Leveling Scan* from the transition piece platform of a turbine at the Global Tech I wind farm.

intersection is a parabola, if its absolute value matches the elevation angle (in our case $3°$), or hyperbola, if the elevation angle is exceeded.

At the point where the laser hits the sea surface, the water absorbs the infrared light and the CNR signal drops significantly. This drop does not occur in the form of a step, since the lidar cumulates measurements over a probe volume, i.e. the measurement for a certain target range consists of a weighted average of measurements around the target distance due to the length and shape of the laser pulses.

Figure 4 shows an example of the CNR value for a sea surface measurement over the range $r_{\mathrm{lidar}}$ for a single beam with the corresponding azimtuh angle $\gamma_{\mathrm{lidar}} \in \Gamma$. $\Gamma$ is the set of viable azimuth angles, i.e. azimuth angles where the sea surface could be detected.

The figure also shows an inverse sigmoid function, which was fitted to the CNR values, in black and the midpoint of the sigmoid function is shown by the vertical line in blue, which is our estimate for the sea surface distance. For the sigmoid function we use a logistic function given by

$$s_{p_{\mathrm{high}}, p_{\mathrm{low}}, p_{\mathrm{midpoint}}, p_{\mathrm{growth}}}(r) = p_{\mathrm{low}} + \frac{p_{\mathrm{high}} - p_{\mathrm{low}}}{1 + \exp\left(p_{\mathrm{growth}}\left(r - p_{\mathrm{midpoint}}\right)\right)}. \tag{8}$$

This function is defined over the range $r$ and is parameterized by $p_{\mathrm{high}}$ and $p_{\mathrm{low}}$ for the upper and lower limit of the sigmoid, respectively, $p_{\mathrm{midpoint}}$ for the sigmoid's midpoint and $p_{\mathrm{growth}}$ for the logistic growth rate. The sigmoid function is fitted to the measured CNR values using a least-squares fit, and the estimate for the sea surface distance is set to

$$r_{\mathrm{sea}} := p_{\mathrm{midpoint}}. \tag{9}$$

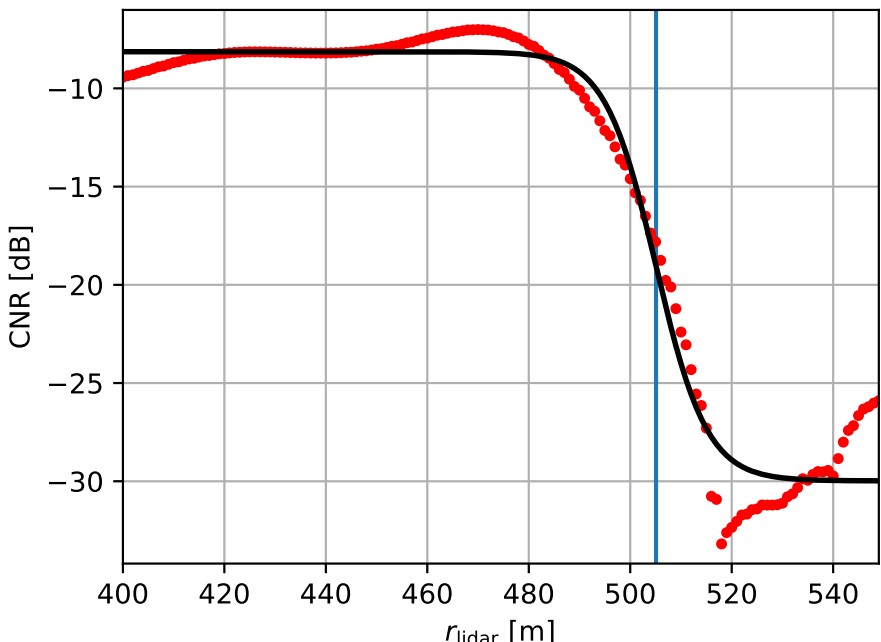

**Figure 4.** Example of a measured CNR over the range for one azimuth angle. The black graph is an inverse sigmoid function fitted to the CNR values and the vertical blue line is in the sigmoid's midpoint, representing the estimated distance to the sea surface.

Depending on the lidar instrument used and the settings, it is possible that the shape of the CNR curve will look different. In our study, the inverse sigmoid fit was found to give a robust estimate of the distance to the water surface, for other systems it may be possible that the fit function needs to be adjusted.

We only considered scans where the CNR value exceeded a defined threshold. If the maximum CNR value for a single beam was below $-18$ dB for all ranges, we omitted that azimuth angle. If less than 20 azimuth angles met these requirements, we removed the entire *Sea Surface Leveling Scan*.

### 2.2.2 Sea Surface Distance Model

To estimate the leveling of the lidar with respect to the sea surface, we built a model to calculate the distances for every azimuth angle to the surface plane for given roll and pitch angles and a height above the surface.

Figure 5 shows the definition of the coordinate system for the lidar as well as the roll angle $\rho$, the pitch angle $\phi$ and the azimuth angle $\gamma$. Note that in this case we have defined the rotations around the axes in a clockwise direction. The following matrices $\left(R_x(\alpha), R_y(\alpha), R_z(\alpha) \in \mathbb{R}^{3 \times 3}\right)$ define these clockwise rotations around the three axis for the substitute angle $\alpha$:



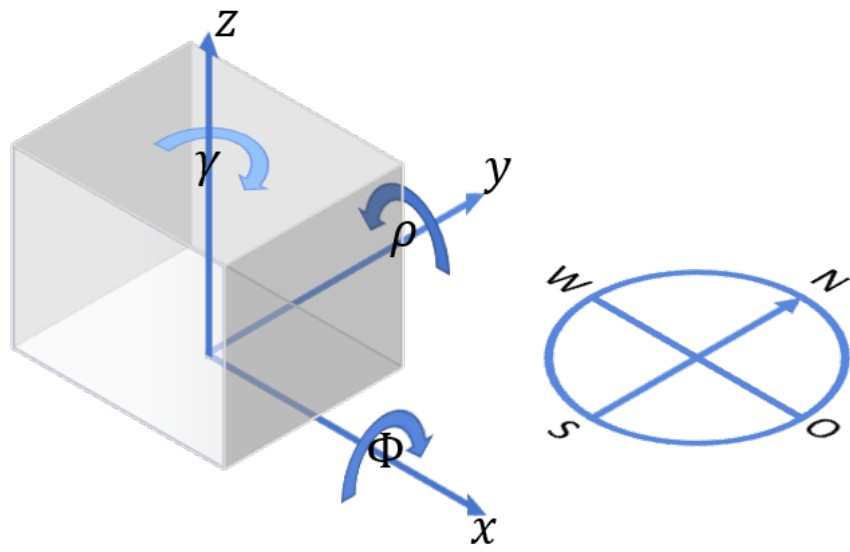

**Figure 5.** Coordinate system of the lidar with the clockwise rotations around the axes and a compass rose. $\gamma$ is the yaw angle, $\rho$ is the roll angle and $\phi$ is the pitch angle of the device.

$$R_x(\alpha) = \begin{pmatrix} 1 & 0 & 0 \\ 0 & \cos(\alpha) & \sin(\alpha) \\ 0 & -\sin(\alpha) & \cos(\alpha) \end{pmatrix}, \quad R_y(\alpha) = \begin{pmatrix} \cos(\alpha) & 0 & -\sin(\alpha) \\ 0 & 1 & 0 \\ \sin(\alpha) & 0 & \cos(\alpha) \end{pmatrix}, \quad R_z(\alpha) = \begin{pmatrix} \cos(\alpha) & \sin(\alpha) & 0 \\ -\sin(\alpha) & \cos(\alpha) & 0 \\ 0 & 0 & 1 \end{pmatrix} \tag{10}$$

Now we can define the building blocks for modelling the laser beam of the lidar. Without any rotations the laser beam $L_{\mathrm{scanner}} \in$
$\mathbb{R}^3$ can be described as a 3D straight line pointing to the north along the $y$-axis by the vector equation:

$$L_{\mathrm{scanner}}(r) := \begin{pmatrix} 0 \\ r \\ 0 \end{pmatrix}, \tag{11}$$

where $r \in \mathbb{R}_{\geq 0}$ is the variable for the range.

We define the pitch $R_{\mathrm{pitch}}(\phi) \in \mathbb{R}^{3 \times 3}$ with the pitch angle $\phi \in [-180°, 180°[$ as:

$$R_{\mathrm{pitch}}(\phi) := R_x(\phi) \tag{12}$$

The roll $R_{\mathrm{roll}}(\rho) \in \mathbb{R}^{3 \times 3}$, with the roll angle $\rho \in [-180°, 180°[$ is defined as:

$$R_{\mathrm{roll}}(\rho) := R_y(\rho) \tag{13}$$



The azimuth $R_{\mathrm{yaw}}(\gamma) \in \mathbb{R}^{3\times3}$ with the azimuth angle $\gamma \in [0°, 360°[$ of the scanner head is given by:

$$R_{\mathrm{yaw}}(\gamma) := R_z(\gamma) \tag{14}$$

We define the elevation $R_{\mathrm{elevation}}(\varepsilon) \in \mathbb{R}^{3\times3}$ with the elevation angle $\varepsilon \in [-180°, 180°[$ of the lidar counter-clockwise around the $x$-axis, therefore we have to adjust the sign:

$$R_{\mathrm{elevation}}(\varepsilon) := R_x(-\varepsilon) \tag{15}$$

And with the height $h \in \mathbb{R}_{\geq 0}$ of the scanner head above the sea surface we can declare the height vector $H \in \mathbb{R}^3$:

$$H(h) := \begin{pmatrix} 0 \\ 0 \\ h \end{pmatrix} \tag{16}$$

In addition, we can add the displacement vector $D \in \mathbb{R}^3$, which consist of $x_{\mathrm{shift}} \in \mathbb{R}$ and the $y_{\mathrm{shift}} \in \mathbb{R}$ the shifts along the $x$- and $y$-axes respectively.

$$D(x_{\mathrm{shift}}, y_{\mathrm{shift}}) := \begin{pmatrix} x_{\mathrm{shift}} \\ y_{\mathrm{shift}} \\ 0 \end{pmatrix} \tag{17}$$

For the Windcube 200S and most other long-range lidar models the laser does not exit from the center of the rotation around the $z$-axis, since the laser is deflected by two mirrors. We did not consider this displacement in our previous work in (Rott et al., 2017). Adding the displacement means that the model no longer describes a conic section and becomes more complex. However, the effects of this change are only marginal for a relatively small displacement as it is the case for a typical two-mirror lidar scanner. For this device we approximated the displacement as $x_{\mathrm{shift}} = -0.15$ m, $y_{\mathrm{shift}} = 0.15$ m. For the sake of completeness, we list the formulas without and with the displacement and use the complete formula for our evaluation. The coordinates of the laser beam $L = L(h, \phi, \rho, \gamma, \varepsilon, r, x_{\mathrm{shift}}, y_{\mathrm{shift}}) \in \mathbb{R}^3$ in the coordinate system of the lidar can now be described by adding the introduced displacements and rotations to $L_{\mathrm{scanner}}$ in the following manner:

$$L(h, \phi, \rho, \gamma, \varepsilon, r, x_{\mathrm{shift}}, y_{\mathrm{shift}}) := H(h) + R_{\mathrm{pitch}}(\phi) \cdot R_{\mathrm{roll}}(\rho) \cdot R_{\mathrm{yaw}}(\gamma) \cdot (R_{\mathrm{ele}}(\varepsilon) \cdot L_{\mathrm{scanner}}(r) + D(x_{\mathrm{shift}}, y_{\mathrm{shift}})) \tag{18}$$

We want to estimate the range $r_0 \in \mathbb{R}_{\geq 0}$ at which the laser beam hits the sea surface i.e. $L(h, \phi, \rho, \gamma, \varepsilon, r_0, x_{\mathrm{shift}}, y_{\mathrm{shift}}) = \begin{pmatrix} x_{\mathrm{L}} \\ y_{\mathrm{L}} \\ 0 \end{pmatrix}$ for the parameters $h, \phi, \rho, \gamma, \varepsilon, x_{\mathrm{shift}}, y_{\mathrm{shift}}$. To do so, we can solve the vector equation for the variable $r_0$ in the $z$-coordinate. While the variable $r_{\mathrm{sea}}$ (see Equation (9)) defines the measured distance to the sea surface we introduce here the variable $r_0$ of the modelled distance as function of the different rotation angles and translational shifts. In our previous publication (cf. Rott et al. (2017)), we set $x_{\mathrm{shift}}, y_{\mathrm{shift}} = 0$:

$$r_0(h, \phi, \rho, \gamma, \varepsilon) = \frac{h}{\cos(\varepsilon)\left(\cos(\gamma)\sin(\phi) - \cos(\phi)\sin(\rho)\sin(\gamma)\right) - \cos(\phi)\cos(\rho)\sin(\varepsilon)} \tag{19}$$



eawe
european academy of wind energy

WIND
ENERGY
SCIENCE
DISCUSSIONS

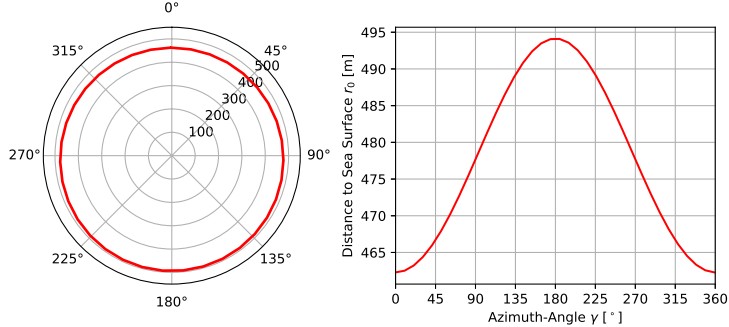

**Figure 6.** Modelled distances to the sea surface for an exemplary set of parameters ($h = 25$ m, $\phi = 0.1°$, $\rho = 0.0°$, $\gamma \in [0, 360[$, $\varepsilon_{\mathrm{ssl}} = -3°$). On a polar plot on the left and a regular plot on the right.

When we include the non trivial displacement $D(x_{\mathrm{shift}}, y_{\mathrm{shift}})$ of the scanner head from the $z$-axis, the complete equation for the range $r_0$ is:

$$r_0(h, \phi, \rho, \gamma, \varepsilon) = \frac{h + x_{\mathrm{shift}} \sin(\gamma) \sin(\phi) + x_{\mathrm{shift}} \sin(\rho) \cos(\gamma) \cos(\phi) + y_{\mathrm{shift}} \sin(\gamma) \sin(\rho) \cos(\phi) - y_{\mathrm{shift}} \sin(\phi) \cos(\gamma)}{\cos(\varepsilon) \left( \cos(\gamma) \sin(\phi) - \cos(\phi) \sin(\rho) \sin(\gamma) \right) - \cos(\phi) \cos(\rho) \sin(\varepsilon)}$$

(20)

Given the set of parameters $h, \phi, \rho, \gamma, \varepsilon, x_{\mathrm{shift}}$ and $y_{\mathrm{shift}}$ we can calculate the range $r_0$ to the sea surface.

Figure 6 illustrates the distance from the lidar to the sea surface for:

$h = 25$ m, $\phi = 0.1°$, $\rho = 0.0°$, $\gamma \in [0, 360[$, $\varepsilon_{\mathrm{ssl}} = -3°$, $x_{\mathrm{shift}} = -0.15$ m, $y_{\mathrm{shift}} = 0.15$ m.

### 2.2.3 Model Fitting

To estimate the leveling of the lidar from the measured *Sea Surface Leveling Scan*, we have to first determine the ranges $r_{\mathrm{sea}}(\gamma)$ to the sea surface for every viable azimuth angle $\gamma \in \Gamma$ (see Section 2.2.1). Then we fit the unknown parameters $(h, \phi, \rho)$ of our model from Equation (20) to these measurements.

To do this we build an optimization problem, which minimizes the deviation of the model to the measurement data. For the cost-function we used the maximum-likelihood method by utilizing the Lorentz-Distribution. The Lorentz-Distribution does not weight outliers as strongly as the commonly used Gaussian distribution, i.e. the least squares method. The cost function is:

$$C_{\mathrm{ssl}}(\phi, \rho, h) \quad := \quad \sum_{\gamma \in \Gamma} \log \left( 1 + 0.5 \left( r_{\mathrm{sea}}(\gamma) - r_0(\phi, \rho, h, \gamma, \varepsilon_{\mathrm{ssl}}, x_{\mathrm{shift}}, y_{\mathrm{shift}}) \right)^2 \right),$$ (21)

with :

$$\varepsilon_{\mathrm{ssl}} \quad = \quad -3°,$$ (22)

$$x_{\mathrm{shift}} \quad = \quad -0.15 \text{ m},$$ (23)

$$y_{\mathrm{shift}} \quad = \quad 0.15 \text{ m}$$ (24)





The set $\Gamma$ is the set of all viable azimuth-angles. Hereby the optimization problem is the following:

$$\min_{\phi,\rho,h} \quad C_{\text{ssl}}(\phi,\rho,h) \tag{25}$$

$$\text{s.t.:} \quad h \in \mathbf{R}_{>0}, \quad \phi,\rho \in [-180°, 180°[ \tag{26}$$

Analogous to the previous optimization problem, we again used the Nelder-Mead method (Gao and Han, 2012) to solve this optimization.

## 2.3 Platform Tilt Model


The PTM estimates the correlation between the turbine operation and the bending of the transition piece platform. The procedure is summarized in the green boxes in Figure 1.

We observed that the platform with the lidar was tilted, depending on the turbine operation. To consider this we assume in accordance with elastic bending of a cantilever beam a linear rotational spring stiffness $k_{\text{tilt}}$.

$$\tau = k_{\text{tilt}} \cdot T, \tag{27}$$

where $\tau$ is the tilt angle. The rotor thrust $T$ and power $P$ are commonly expressed as function of the wind speed $u$, the swept rotor area $A$, air density $\rho$ and the dimensionless thrust and power coefficient, $c_{\text{T}}$ and $c_{\text{P}}$, resprectivly:

$$T = c_{\text{T}}(u)\frac{\rho}{2}Au^2 \tag{28}$$

$$P = c_{\text{P}}(u)\frac{\rho}{2}Au^3 \tag{29}$$

Combining Equation (27) to (29) yields the tilt angle as function of the power $P$ and the wind speed $u$ representing temporal averages for the time during which the *Sea Surface Leveling Scan* was executed.

$$\tau = \frac{k_{\text{tilt}} \cdot c_{\text{T}}(u)}{c_{\text{P}}(u)}\frac{P}{u} \tag{30}$$

For most wind turbines the thrust and power coefficient can be approximated as constant in the variable operational speed range which corresponds to the lower to medium partial load range. Beyond and especially above the rated wind speed the

dependency on the wind speed has to be considered since the thrust loading is reduced while the rated power is maintained. In the following we consider only the variable operational speed range, typically extending between $3.5 \text{ ms}^{-1}$ and $9 \text{ ms}^{-1}$ to $11 \text{ ms}^{-1}$, to simplify Equation 27 with a constant factor $c$.

$$\tau = c \cdot \frac{P}{u} \tag{31}$$

To define the tilt of the platform we are using the rotational matrices we defined in Section 2.2.2. The tilt matrix $R_{\text{tilt}}$ of the

platform depends on the tilt angle $\tau$ and the direction $\nu$ in which the platform tilts (in our case this is the nacelle orientation). We define:

$$R_{\text{tilt}}(\tau,\nu) := R_z(\nu)R_x(-\tau)R_z(-\nu) \tag{32}$$





The negative tilt angle $\tau$ has to be considered for the rotation $R_x$, because the platform bends in the negative direction of the turbine orientation $\nu$. This allows to model the rotation of the platform as a combination of resting pitch and roll angles $\phi_{\rm r}$ and

$\rho_{\rm r}$ and the thrust-dependent tilt rotation of the platform.

$$\varepsilon_i = R_{\rm tilt}(\tau_{\rm m,i}, \nu_{\rm m,i})R_{\rm pitch}(\phi_{\rm r})R_{\rm roll}(\rho_{\rm r}) - R_{\rm pitch}(\phi_{\rm m,i})R_{\rm roll}(\rho_{\rm m,i}), \quad \forall i \in \Omega \tag{33}$$

The matrix $\varepsilon_i$ is the $i$-th residual of the combined rotations measured by the SSL in terms of the pitch angle $\phi_{\rm m,i}$ and the roll angle $\rho_{\rm m,i}$, and the modelled rotations defined by the corresponding tilt angle $\tau_{\rm m,i}$ and orientation $\nu_{\rm m,i}$ of the turbine. While $\Omega$ is the set of all available measurements at different operating conditions. When we substitute $\tau_{\rm m,i}$ with the measured active

power $P_{\rm m,i}$ and wind speed $u_{\rm m,i}$ by using Equation (31), we are left with the three unknown parameters: $c, \phi_{\rm r}, \rho_{\rm r}$. To estimate these parameters we try to minimize the residual $\varepsilon_i$ with the following cost function:

$$C_{\rm tilt}(c, \phi_{\rm r}, \rho_{\rm r}) := \sum_{i \in \Omega} \left\| R_{\rm tilt}\left(c \cdot \frac{P_{\rm m,i}}{u_{\rm m,i}}, \nu_{\rm m,i}\right) R_{\rm pitch}(\phi_{\rm r})R_{\rm roll}(\rho_{\rm r}) - R_{\rm pitch}(\phi_{\rm m,i})R_{\rm roll}(\rho_{\rm m,i}) \right\|_2^2 \tag{34}$$

The operator $\|\cdot\|_2$ is the Frobenius norm. The optimization problem, utilizing the least-square method then is:

$$\min_{c, \phi_{\rm r}, \rho_{\rm r}} \quad C_{\rm tilt}(c, \phi_{\rm r}, \rho_{\rm r}) \tag{35}$$

$$\text{s.t.:} \quad c \geq 0, \quad \phi_r, \rho_r \in [-180°, 180°[ \tag{36}$$

Again, we solved this optimization with the Nelder-Mead algorithm (Gao and Han, 2012). For the initial solution we chose: $c_{\rm init} = 0, \phi_{\rm init} = 0°, \rho_{\rm init} = 0°$. Thus, we can determine the parameters from the measured data.

In the last step we want to utilize the obtained parameters $c, \rho_r, \phi_r$ to estimate the dynamic roll and pitch angle from the power and wind speed measurements of the turbine. This makes it possible to perform further lidar scans and estimate the leveling

and the resulting height and position error of the lidar measurement based on the current SCADA data in post-processing. To do so we are using Equation (33) and assume that the residual $\varepsilon_i$ is zero. We rename the variables $\phi_{\rm m,i}$ and $\rho_{\rm m,i}$ for the pitch and roll angle measured by the SSL to $\phi_{\rm WT,i}$ and $\rho_{\rm WT,i}$, respectively, to emphasize that in this case the variables are derived from the wind turbine data.

$$R_{\rm pitch}(\phi_{\rm WT,i})R_{\rm roll}(\rho_{\rm WT,i}) = R_{\rm tilt}(\tau_{\rm m,i}, \nu_{\rm m,i})R_{\rm pitch}(\phi_{\rm r})R_{\rm roll}(\rho_{\rm r}), \quad \forall i \in \Omega \tag{37}$$

The objective is to solve for $\phi_{\rm WT,i}$ and $\rho_{\rm WT,i}$. To do this, we multiply Equation (37) with the $z$-unit vector $\boldsymbol{e}_z := (0,0,1)^{\rm T}$. Thus, the two sides of the equation represent how the rotation matrices affect the lot vector to the $xy$-plane, resulting in a unit vector, which is orthogonal to the sea surface $\boldsymbol{n}$. With this we get:

$$\begin{pmatrix} -\sin(\rho_{\rm WT,i}) \\ \sin(\phi_{\rm WT,i})\cos(\rho_{\rm WT,i}) \\ \cos(\phi_{\rm WT,i})\cos(\rho_{\rm WT,i}) \end{pmatrix} = \boldsymbol{n}, \quad \text{with} \quad \boldsymbol{n} = R_{\rm tilt}(\tau_{\rm m,i}, \nu_{\rm m,i})R_{\rm pitch}(\phi_{\rm r})R_{\rm roll}(\rho_{\rm r})\boldsymbol{e}_z \tag{38}$$

This can be reformulated to:

$$\rho_{\rm WT,i} = \arcsin(-\boldsymbol{n}_{x_i}), \quad \phi_{\rm WT,i} = \arcsin\left(\frac{\boldsymbol{n}_{y_i}}{\cos(\arcsin(-\boldsymbol{n}_{x_i}))}\right). \tag{39}$$

Now we can derive the dynamic roll and pitch angles from the SCADA data.





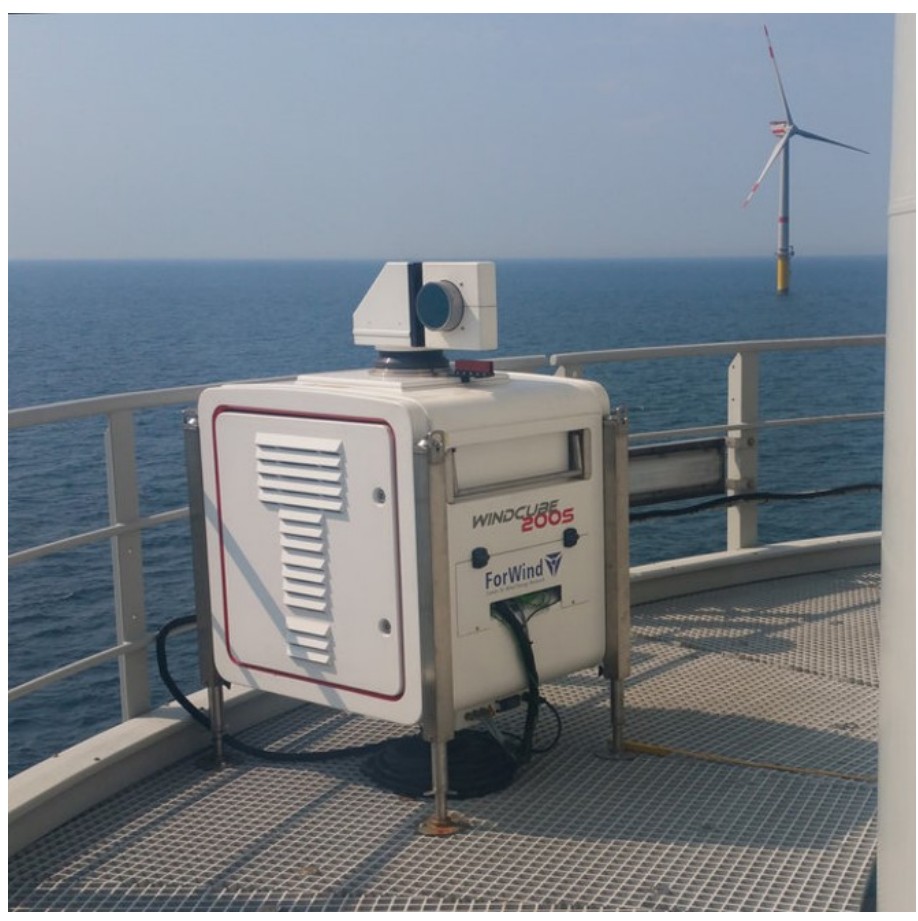

**Figure 7.** Image of the WindCube200S installed on the transition piece of the wind turbine in the offshore windfarm Global Tech I. Picture taken by Stephan Voß.

## 2.4 Measurement set-up and data

The three methods presented in this paper refer to a measurement campaign which was carried out from August 2018 until June 2019 in the German North Sea with a scanning lidar positioned on the platform of the transition piece of a wind turbine in the offshore wind farm Global Tech I (GT I). A more detailed description of the measurement campaign is given in (Schneemann et al., 2020a), some measurement data is published by Schneemann et al. (2019). In the following sections we will first present important information about the measurement setup, equipment and the wind farm GT I (Section 2.4).

The offshore wind farm GT I consists of 80 turbines of the type Adwen AD 5-116 with a total nominal power output of 400 MW. It is located in the German North Sea approximately 100 km from the coast.





For the measurement campaign we had knowledge about the coordinates of all turbines as well as a subset of 1 Hz Supervisory Control and Data Acquisition (SCADA) data, including active and reactive power output, nacelle orientation, wind speed, wind direction and turbine status.

The scanning long-range Doppler lidar we used for this investigation is a Leosphere WindCube 200S (serial number WLS200S-024) (see Figure 7). It was installed on the transition piece platform of the turbine GT58 at the westerly border of the offshore wind farm GT I. For detailed information about the lidar, we refer to Schneemann et al. (2020a).

## 3   Results

In the following the results of the HT, the SSL and the PTM are presented for the reference case.

### 3.1   Hard Targeting

Since the lidar was installed on the platform southwest of the tower of wind turbine GT58, we could not see a large part of the wind farm GTI from the lidar. But the turbines north and southeast were visible. When we installed the lidar it was oriented south-south-east. We made a rough estimate for the lidar's uncorrected north orientation of $\gamma_{\mathrm{init}} = 170°$ and used this as our initial estimate. We defined the coordinates of the turbine GT58 as the origin of our coordinate system and used its coordinates as the initial guess for the location of the lidar i.e. $x_{\mathrm{init}} = y_{\mathrm{init}} = 0$ m.

Figure 8 presents the results of the optimization for the whole set of available hard targets that could be detected from the lidar's position. The orange dots represent the identified hard targets. The blue symbols show the positions of the turbines. The optimization yielded $x_{0,\mathrm{opt}} = -3.9$ m, $y_{0,\mathrm{opt}} = -4.3$ m and $\gamma_{0,\mathrm{opt}} = 171.65°$.

For comparison, we applied HT not only to the entire set of detected WTs, but also to the individual WTs (Scenario 1: "individual turbines"), a subset of the various WTs located up to a certain range away from the lidar (Scenario 2: "increasing range"), and a subdivision of the WTs into northern and southern WTs from the lidar (Scenario 3: "north/south"). These three additional scenarios allowed us to observe the sensitivity of the method with respect to the set of available hard targets and their distances.

In the first scenario we applied the HT to the 17 individual turbines, which could be detected from the lidar's point of view (see Figure 8. In this case one has to be careful that the algorithm assigns the hard targets to the correct WT, since the optimization can easily converge against a wrong WT if the initial values are not ideal. On average, this resulted in a lidar alignment of $171.71°$ with a standard deviation of $0.58°$.

In the second scenario, we artificially limited the range of the lidar. We considered six cases where we increased the range in 1000 m increments from 1000 m to 6000 m, with the goal of investigating how the results change with the addition of more hard targets.

Figure 9 shows the evolution of position determination and alignment over the range. On average, the alignment was $171.63°$ with a standard deviation of $0.02°$.



WIND
ENERGY
SCIENCE
DISCUSSIONS

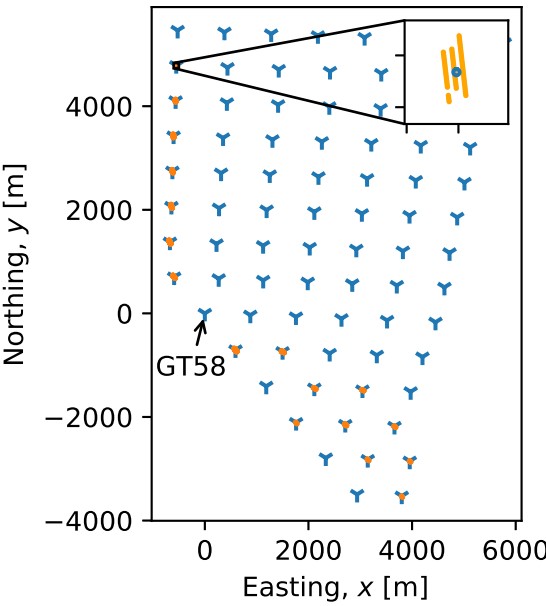

**Figure 8.** Result of the HT. The blue symbols represent the turbine layout. The orange dots are the hard target measurements $L_{\mathrm{ht}}$. In the upper right corner is a zoom in for the northernmost hard target measurements.

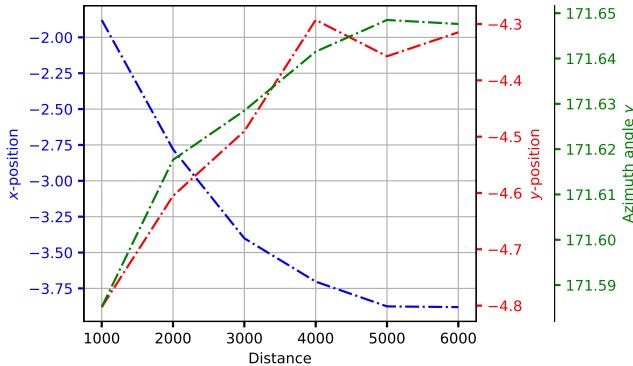

**Figure 9.** Result of the HT for increasing range (scenario 2). The blue graph shows the evolution of the $x$-coordinate over the range restriction. The red graph shows the $y$-coordinate and the green graph the azimuth angle, respectively.

In the third scenario, we divided the hard targets into two groups. Those from the lidar to the north and the southern hard targets. The mean value for the alignment from these two groups is $171.65°$ with a standard deviation of $0.03°$.

In Table 1 the statistics for the three scenarios are summarized.





**Table 1.** Results of the HT for three scenarios.

| scenario | statistic | $x$-position | $y$-position | azimuth angle $\gamma$ |
|---|---|---|---|---|
| 1. individual turbines | mean | 8.76 m | 10.27 m | 171.71° |
|  | std | 9.04 m | 12.17 m | 0.58° |
| 2. increasing range | mean | -3.25 m | -4.48 m | 171.63° |
|  | std | 0.79 m | 0.20 m | 0.024° |
| 3. north/south | mean | -4.63 m | –2.33 m | 171.65° |
|  | std | 4.58 m | 5.84 m | 0.027° |

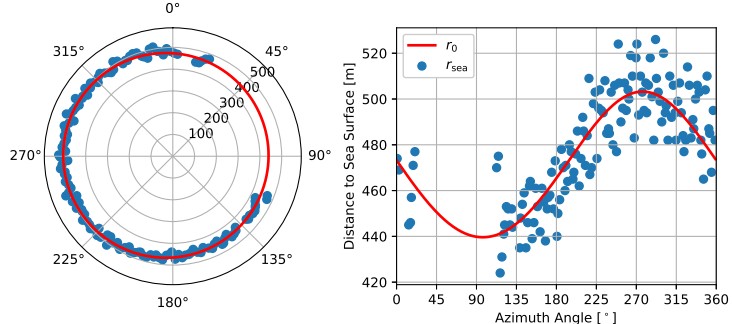

**Figure 10.** Exemplary fitting of the modelled sea surface distances to the measured distances for a complete SSL scan. $\phi_{\mathrm{ssl}} = -0.025°, \rho_{\mathrm{ssl}} = -0.201°$ and $h_{\mathrm{ssl}} = 24.560$ m. Illustrated in a polar plot on the left and in a regular plot on the right.

## 3.2 Sea Surface Leveling

### 3.2.1 Lidar Leveling

In the first step, we determined the intersections of the laser beam of the lidar with the sea surface for the *Sea Surface Leveling Scans* according to Section 2.2.1. With the Sea Surface Distance Model (Section 2.2.2) we can use the optimization from Section 2.2.3 to find the optimal set of parameters for the height, roll angle and pitch angle. For the initial guess for our parameters we chose $h_{\mathrm{init}} = 25$ m, $\phi_{\mathrm{init}} = 0°, \rho_{\mathrm{init}} = 0°$.

Figure 10 illustrates an exemplary result of the model fitting for a single SSL scan. The blue dots show estimated distances to the sea surface, which are scattered around the model fit shown in red. Some uncertainty can be seen in the measurements, but the fit still seems to reflect the distances well. The optimization for this scan yielded $\phi_{\mathrm{ssl}} = -0.025°, \rho_{\mathrm{ssl}} = -0.201°$ and $h_{\mathrm{ssl}} = 24.560$ m.



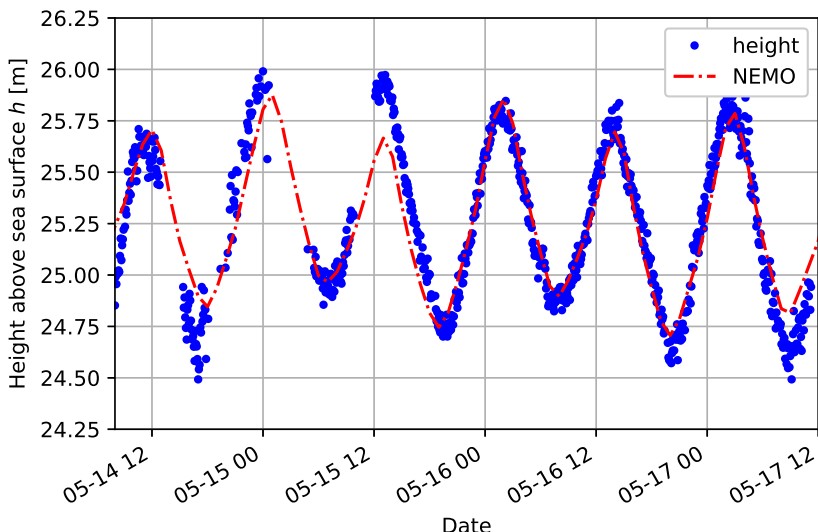

**Figure 11.** Height above sea level determined during three days of continuous measurements. The blue dots represent the height of the lidar's scanning head above the sea surface as measured by SSL for a contiguous subset of the SSL scan data. Offset-corrected sea surface height from the NEMO model provided by the E.U. Copernicus Marine Service is shown in red.

We repeated this procedure for every available SSL scan from our measurement campaign (see Section 2.2.1). On a standard
personal computer, the optimization took about 1 to 2 seconds to compute a suitable solution per scan. Of our measurements, 1935 scans passed our filter requirements, for which we applied the SSL.

Figure 11 and Figure 12 display the results of our method for a consecutive subset of our measurements. Figure 11 reflects the tidal pattern in the German North Sea. As reference we also show offset-corrected *Nucleus for European Modelling of the Ocean* (NEMO) data (red graph) from the E.U. Copernicus Marine Service (Madec and Team). In Figure 12 we can see the
roll $\rho_{\mathrm{m}}$ and pitch $\phi_{\mathrm{m}}$ angle of the lidar, which describe the leveling of the device, over the same time period. While consecutive measurements result in very similar values for the angles, over the course of the time period we can observe significant changes of the angles, which we attribute to the variable thrust loading of the turbine at different wind speeds.

### 3.2.2 Platform Tilt Angles

To easily process the results of the SSL along with the SCADA data needed to determine the platform tilt, we resampled the
data to 5-minute time steps using 5-minute averages. With this we have a joined set of 1142 measurements. After fitting the PTM to the measurements we achieved the following results for the three parameters:

$$c = 3.5° \cdot 10^{-4} \; \frac{\mathrm{m}}{\mathrm{s} \cdot \mathrm{kW}}, \quad \phi_{\mathrm{r}} = 0.025°, \quad \rho_{\mathrm{r}} = -0.109° \tag{40}$$

The parameter $c$ allows us to estimate the tilt angle of the transition piece platform according to Equation (31).



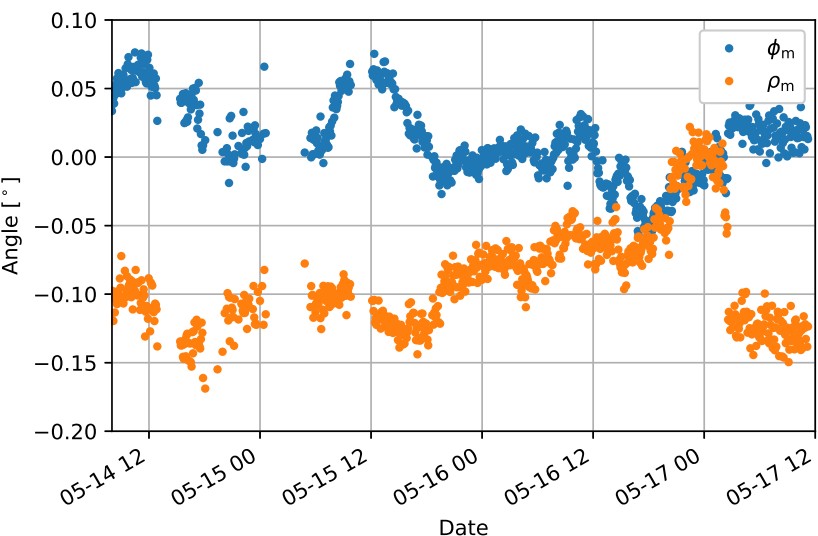

**Figure 12.** Roll $\rho_\mathrm{m}$ and pitch $\phi_\mathrm{m}$ angle of the lidar measured by the SSL for a contiguous subset of the SSL scan data.

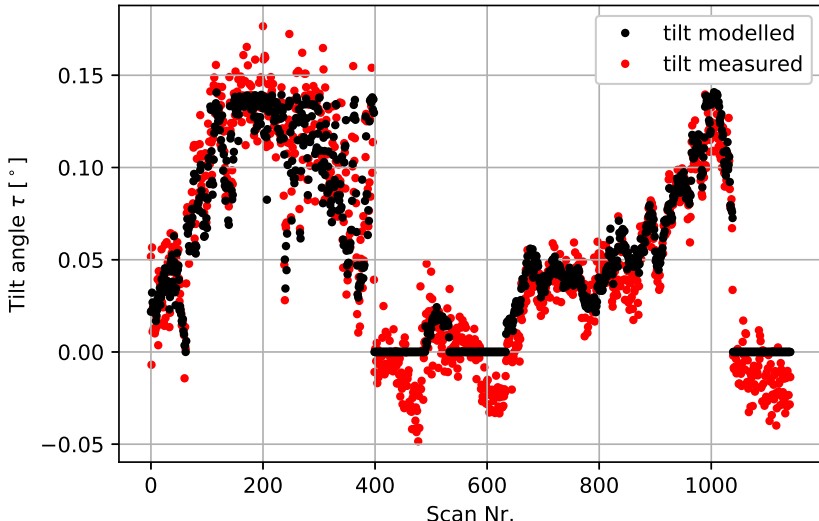

**Figure 13.** Comparison of the tilt angle $\tau_\mathrm{m}$ of the transition piece platform estimated by the PTM in black and the tilt angle derived from the roll $\rho_\mathrm{m}$ and pitch $\phi$m angle measured by the SSL over the scan number.

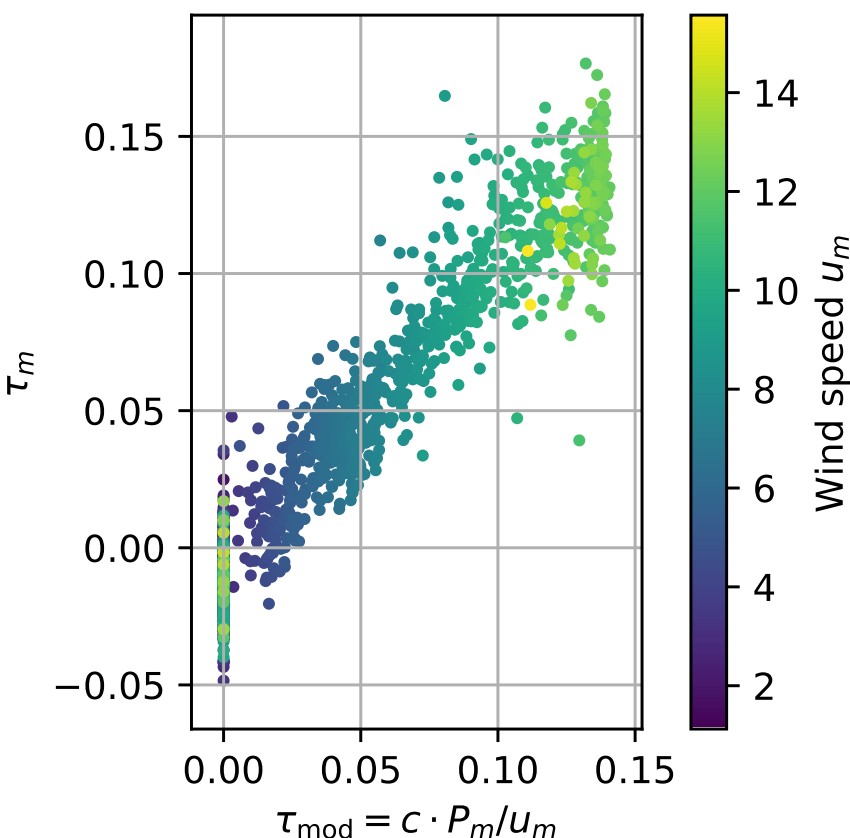

**Figure 14.** Comparison of the tilt angle of the transition piece platform estimated by the PTM ($x$-axis) to the tilt angle derived from the roll $\rho_\mathrm{m}$ and pitch $\phi$m angle measured by the SSL ($y$-axis) in a scatter plot. The color of the scatter points represent the wind speed measured by the tubine anemometer.

In Figure 13 a comparison of the estimated tilt angles is depicted for all joined measurements. The black dots show the
tilt angle calculated by the PTM with the SCADA data and the parameter $c$, we refer to as modelled tilt angle. The red dots
show the tilt angle of the lidar derived from the roll $\rho_\mathrm{m}$ and pitch $\phi$m angles measured by the SSL using Equation (33) and
the estimated resting roll $\phi_\mathrm{r}$ and pitch angle $\rho_\mathrm{r}$, respectivly, by minimizing the residual $\varepsilon_i$. We refer to this as the measured tilt
angle. The maximum modelled tilt angle is around $0.14°$, when the turbine is in normal operation and the wind speed is close
to the nominal wind speed. There are a few periods, where the estimation of the PTM yields $0°$. They coincide with situations
without power production and associated thrust loading. In these situations the measured tilt angle drops below $0°$. We believe
that the weight of the rotor pulls the tower and consequently the transition piece platform forward, creating a negative tilt angle.
However, the amount is relatively small. Therefore, we have not refined the model in this respect.

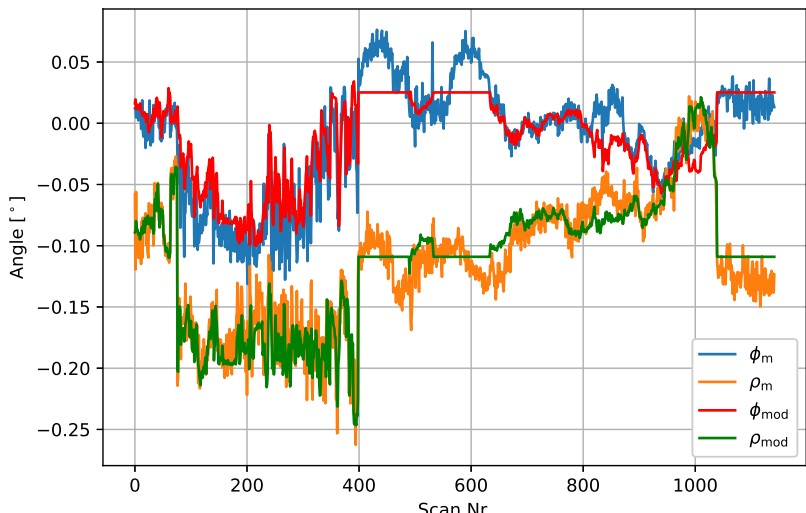

**Figure 15.** Comparison of the roll $\rho_{\mathrm{m}}$ and pitch $\phi_{\mathrm{m}}$ angle measured by the SSL and the roll $\rho_{\mathrm{WT}}$ and pitch angle $\phi_{\mathrm{WT}}$ modelled by the PTM

**Table 2.** Error statistics of the modelled roll and pitch angles to the measured roll and pitch angles.

| statistic | $(\rho_{\mathrm{m},j} - \rho_{\mathrm{mod},j})_j$ [°] | $(\phi_{\mathrm{m},j} - \phi_{\mathrm{mod},j})_j$ [°] |
|---|---|---|
| count | 1142 | 1142 |
| mean | $4.6 \cdot 10^{-5}$ | $4.8 \cdot 10^{-5}$ |
| RMSE | 0.0174 | 0.0204 |
| min | -0.0607 | -0.0872 |
| 25 % | -0.0117 | -0.0104 |
| 50 % | -0.0002 | -0.0005 |
| 75 % | 0.0112 | 0.0112 |
| max | 0.0977 | 0.0529 |

In Figure 14 the measured tilt angle and the modelled tilt angle are compared in a scatter plot to demonstrate the strong linear correlation between our thrust estimation (see Equation 31) and the tilting of the transition piece platform.

In Figure 15 we can see the modelled roll and pitch angle in comparison to the values measured by the SSL. We obtain them by inserting the estimated parameters into Equation (38) and then using Equation (39). A few instances can be observed where the modelled angles remain temporarily constant at their resting positions $\rho_{\mathrm{r}}$ and $\phi_{\mathrm{r}}$. These are situations where the turbine has not produced any power and thus the thrust in our model has been set to $0$ kN and thereby the tilt angle is also $0°$.





Table 2 summarizes the statistical deviations between the measured and the modelled roll and pitch angles. The table shows
the number of measurements (count), the average deviation (mean), the root mean square error of the estimate (RMSE), the
largest negative deviation between the measured and the modelled angles (min), the lower quartile of the error (25 %), the
median (50 %), the upper quartile (75 %) and the largest position deviation of the measured and the modelled angles (max) for
the roll and the pitch angle respectively.

## 4   Discussion

The *Hard Targeting Scans* and *Sea Surface Leveling Scans* are very valuable measurements for an offshore lidar campaign.
They do not require additional equipment and only take a rather small amount of time, but can help improve the accuracy
of the measurement campaign significantly. The knowledge of the correct orientation and position of the lidar is essential for
measurement campaigns where local effects, e.g. turbine wake, induction zone, are analysed. Information about the leveling of
the device is useful for all measurements, but especially for long-range measurements, since the error in measurement height
due to a misalignment scales linearly with the range.

HT to determine the north alignment and position of a long-range lidar is a commonly used technique that probably evolved
from cross bearing in navigation and is based on standard geometry. Nevertheless, we have not found a consistent description of
the method in the literature that relates to the calibration of long-range Doppler wind lidars and can be easily applied in a wind
farm. Therefore, we wanted to dedicate a small part of this paper to the presentation of a simple method that can accurately
determine the position of the lidar in addition to the north orientation. In our study we applied the method to a subset of the
available hard targets in three different scenarios to investigate the sensitivity of the method towards the amount of suitable
data.

In Scenario 1 we applied the HT to the 17 individual wind turbines we could detect in the *Hard Target Scans* and computed
mean and standard deviation of the orientation and positioning. The results show that the method is not very accurate when
applied to individual turbines. Especially in the case of positioning, there are larger deviations in the results. This is due to
the fact that the detection of individual hard targets is not very accurate because of probe length averaging. In addition, only
the coordinates corresponding to the center of the tower were available for the individual turbines. Unfortunately, no statement
could be made about the accuracy of these coordinates.

In Scenario 2, the method was applied to hard targets within a certain measurement range. The range was successively
increased until all available data was included. This example demonstrates how adding additional hard targets ensures that the
results for positioning and alignment each converge against a constant value.

In the 3rd scenario, the set of available hard targets was divided into the northern and southern turbines, when viewed from
the lidar. The results show that the orientation is determined almost identically for both subsets, but there are differences of
several meters in the positioning. This is due to the fact that for both subsets the vectors to the respective hard targets are almost
linearly dependent on each other. For a more precise positioning, reference points are needed which are located in as different
directions as possible (ideally orthogonal to each other).



Overall, the accuracy of the determination of position and orientation by the method presented here increases with the number of hard targets detected. The results from scenario 2 show that especially the determination of the orientation has a very high accuracy.

The SSL is a rather new technique, which was first introduced in Rott et al. (2017). In this paper we extended the method and explained it in more detail. The advantage of this method is, that by only using the lidar itself and the sea surface, it reaches a very high accuracy in determining the leveling. Since the laser itself is utilized, this is even an advantage over additional alignment sensors attached to the lidar's housing.

Figure 6 shows that a misalignment of $0.1°$ leads to a maximum difference in distance of more than $30$ m for the given
setup. This demonstrates the sensitivity of the shape of the intersection with regards to the pitch angle of the lidar device. This sensitivity strongly depends on the elevation angle of the PPI scan. Here we have chosen an angle of $\varepsilon_{ssl} = -3°$ based on experience from previous campaigns. A smaller absolute angle further increases the sensitivity of the intersection shape, but this also increases the uncertainties caused by waves. In general, the elevation must be chosen to match the measurement setup, the most important thing being that there is a direct line of sight to the water surface for as large a sector as possible. The
setting of the range gates should also be selected carefully. In our case, we chose a $1$ m resolution that sufficiently resolves the drop in CNR value at which the beam hits the water (see Figure 4).

A large source of uncertainty is the determination of the distance to the sea surface (Section 2.2.1). It is possible that the method presented here systematically overestimates or underestimates the distances. However, as long as this is done consistently, only the size but not the general shape of the intersection with the water surface is affected by this. This leads
to an error in the estimation of the height, but not in the estimation of the angles. Due to the flat elevation angle, an over- or underestimation of the distance by e.g. $20$ m results in an over- or underestimation of the height above the water surface by approx. $1$ m. Another source of uncertainty in the determination of the distance to the sea surface are waves, thus the best time for the SSL scan is when the water surface is very calm. However, the effects of a misaligned lidar are so significant that we perceive the influence of the waves as measurement uncertainty, which is largely averaged out in the measurements. A further
source for the deviation of the measured distances and the model fitting, which can be observed in Figure 10 is movement of the transition piece platform during an SSL scan. With our setup one SSL scan took approximately 5 minutes to execute, therefore fluctuations with a higher frequence change the shape of the intersection of the laser beam with the sea surface. This is not considered in the model fitting. The SSL as described here therefore gives an average value for the leveling over the scan duration. Furthermore, the movement of the transition piece platform is only modeled as a rotation at the scanner head. It is
likely that the origin of the rotation for the tilt is located further below, resulting in movement of the lidar. However, since the angles are relatively small, the motion is primarily lateral and does not affect the distances to the sea surface, only the change in height due to bending of the tower base directly affects the distance to the sea surface. Since these changes are very small and, moreover, the point of origin cannot be accurately determined, we have neglected this. Lastly, the accuracy of the model fitting also depends on the quality of the SCADA data. Especially the nacelle orientation has an important role for the thrust model.
A deviation in the measurement of the nacelle orientation leads to a systematic error in the tilt model. For our measurement campaign we checked this sensor and assumed that the error or the orientation of the turbine is within $2°$.



Nevertheless, the results of the SSL are considered very reliable. This is supported by the fact that successive measurements for leveling lead to very similar results, which can be seen in Figure 12. In addition, the temporal development of the roll and pitch angle can be described on average very well by means of the PTM (see Figure 15). If we look at the results in Table 2, we can see that the RMSE is only about $0.02°$ for both roll and pitch. We assume that the total model error is of the same order of magnitude. Without power output the turbine's thrust model yields zero loading and the modelled angles take their resting position. In Figure 15 we can observe periods where this occurs and the modelled roll and pitch angle are constant. In these situations we can clearly see a deviation to the measured values and therefore the thrust model does not seem to work that well. This is probably because a turbine also exerts wind loading when it is not producing power and even if the wind speed and therefore the thrust are very low, the tower top tilt moment due to the excentric weight of the rotor-nacelle assembly can lead to an inclination of the transition piece platform, which is not considered in our tilt model. In this respect, the model could be improved in the future.

SSL allows to determine the leveling of a lidar. Thereby larger inclinations can be identified and corrected at the beginning of a measurement campaign. But even if a correction is not possible for logistic reasons and, as we discovered, dynamic variations of the levelling occur, it is helpful to know the exact orientation of a lidar so that the measurements can be interpreted accordingly. To illustrate, we consider an example of the effects that inclinations can have on measurements. In our measurement campaign we observed a maximum absolute roll angle of about $0.25°$ (see Figure 15). This angle alone, for a horizontal PPI scan, causes the height of measurements in 1 km distance to be wrong by up to approx. 4.4 m depending on the azimuth angle. For typical long-range lidars measuring up to 8 km, this builds up to about 35 m, which is more than the height above the water surface of about 25 m for our measurement campaign. This means that at a distance of about 6.5 km we would have hit the sea surface at a downward inclination (Earth curvature taken into account), while at an upward inclination we would have more than doubled the measurement height.

The transition piece platform in our measurement campaign was mounted on a tripod with a water depth of about 40 m. Even though the water depth is comparatively large, we would expect that even larger tilt angles can occur at wind turbines that are mounted on a more flexible monopile and probably even larger at floating wind turbines. For nacelle-based lidar measurements the movement of the nacelle is in the order of one to two degrees. This emphasizes the importance of knowing the correct alignment. We think that in the future additional sensors should be utilized that can directly measure the leveling of the lidar with high accuracy. These should be compared to the results of the SSL, thus providing a reference for the alignment of the laser to which the correct installation of the sensor can be verified. Furthermore, it should be explored how information from such sensors can be directly incorporated into the lidar control to dynamically correct the elevation of the scanner head, or whether it is possible to mount the lidar or the scanner of the lidar in a Cardan suspension.

# 5 Conclusions

The objectives of this paper were to present two methods for accurately estimating the position, orientation, and leveling of a long-range lidar in an offshore environment, and in addition to introduce a model that can estimate turbine-induced inclinations
from turbine operating data.

HT, SSL and the PTM are described in detail and applied to data from an offshore measurement campaign. For HT, we have shown that the presented method takes advantage of detecting multiple HTs in different directions, thereby increasing the accuracy. While we assume an error of about 1 m for the positioning of the instrument, the northing results in consistent values, which indicate a very high accuracy (see Table 1. For SSL and the PTM, the evaluations (Table 2) show that the inclinations
measured by the SSL can be reproduced by the PTM with a very high accuracy. In addition, the SSL also provides a good estimation of the water surface height.

The HT is not limited to being used offshore and should be applied to every long-range lidar campaign to estimate the orientation and positioning at the start of a measurement campaign. While the PTM presented here was developed specifically for use of a lidar on the platform of a transition piece, it can be extended to other offshore sites where tilt is related to the thrust
of the wind turbine.

In general, knowledge about the positioning, orientation and leveling of remote sensing equipment is crucial for successful measurements at longer distances. We therefore consider it very important to perform calibration measurements as proposed in this paper during each measurement campaign.

*Code and data availability.*   The authors plan to publish the scripts for all calculations as well as subsets from the dataset to reproduce most
of the work presented and for use in future measurement campaigns after the open review is launched.

*Author contributions.*   AR initiated and directed the research; developed the methods, prepared the computational scripts, and analyzed the measured data; was heavily involved in funding acquisition and research discussion; prepared the figures; and was lead author of the article. JS assisted with data provision and data processing, as well as derivation of the methods, and provided extensive feedback in several reviews. FT assisted in the development of the computational scripts and verified the calculations; provided support in countless discussions and
gave extensive feedback in several interactions. JJT supported with his rich experience on lidar systems and was instrumental in drafting the research question and helped with extensive review. MK was instrumental in acquiring funding and supervised the research and provided support with valuable comments and good advice.

*Competing interests.*   The authors declare no conflict of interest.



*Acknowledgements.* We performed the lidar measurements and parts of the work in the framework of the research projects "OWP Control"

(FKZ 0324131A) and "X-Wakes" (FKZ 03EE3008D) both funded by the German Federal Ministry for Economic Affairs and Energy on the basis of a decision by the German Bundestag. The work of Frauke Theuer is supported by the German Federal Environmental Foundation (DBU) (Grant Nr. 20018/582). We acknowledge the wind farm operator Global Tech I Offshore Wind GmbH for providing SCADA data as well as their support of the work and the measurement campaign. Thanks to E.U. Copernicus Marine Service Information for providing sea surface height data from the NEMO model used in Figure 11. Special thanks to Johannes Schulz-Stellenfleth, Helmholtz Zentrum

Geesthacht, for helping with sea surface height data. Many thanks to Thomas Luhmann, Jade University of Applied Sciences for the useful comments and advice. Special thanks to Stephan Voß, ForWind – University Oldenburg for his work and support in planning and conducting the measurement campaign.



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
