# Peer review of "Alignment of scanning lidars in offshore wind farms"

_Wind Energy Science, 2021_

## Author Response (AR1)

**SSL Revision Authors Comments**

**November 2021**

We would like to thank both anonymous referees for their constructive feedback and comments which helped to improve our manuscript. We appreciate the time and efforts they put into the review. In the following we reply to the comments point-by-point.

On behalf of the authors,

Andreas Rott

**1 Introduction**

To make the methods presented here more accessible, we have published the code developed in this research and the data used for the analysis presented in the publication. The reader can download the scripts and run them with the data for illustration. The scripts are written in Python (3.8.12).

The code and scripts developed in this research are published at:

Andreas Rott, Jörge Schneemannand Frauke Theuer, "AndreasRott/Alignment _of_scanning_lidars_in_offshore_wind_farms: Version1.0". Zenodo, Nov. 08, 2021. doi: 10.5281/zenodo.5654919.

The data used for the evaluation can be used as an illustration of the code and can be found here:

Andreas Rott, Jörge Schneemann and Frauke Theuer, "Data supplement for "Alignment of scanning lidars in offshore wind farms" - Wind Energy Science Journal", Wind Energy Science Journal. Zenodo, Nov. 08, 2021. doi: 10.5281/zenodo.5654866.

These references were also added to the sections code availability and data availability of the manuscript.

**2 Referee A**

**2.1 General Comments**

**RC A-1:** This paper focuses on obtaining high-precision positioning, orientation, and leveling of lidar device at the offshore measurement campaign. This is a prerequisite for obtaining accurate lidar wind measurement dataset and clearly understanding or explaining meteorological phenomena of offshore wind energy.

The authors describe Hard Targeting and Sea Surface Leveling to accurately estimate the position, orientation, and leveling of a long-range lidar without additional equipment or sensors. Considering the quasi-static inclination caused by the thrust loading of the wind turbine, this paper presents a Platform Tilt Model. Results are generally very convincing. However, before the manuscript can be recommended for acceptance for publication, I have several suggestions and comments here that need to be addressed.

**AC A-1:** We thank the referee very much for the constructive questions and suggestions for improvement. In the following, we will try to answer the questions as clearly as possible and adapt the manuscript in the relevant places.

**2.2 Special comments**

**RC A-2:** Line 115: Please try to give the explanation of the symbol "=#".

**AC A-2:** The symbol # is used in mathematical set theory to denote the cardinality of a set. This term means the number of elements that are in the set. Since this term is not often used in common language, we extended the text as follows.

> Let $N_{ht} := \# L_{ht}$ be the cardinality of the identified hard target measurements $L_{ht}$, i. e. $N_{ht}$ is the number of elements in the set $L_{ht}$.

**RC A-3:** Line 164-165: Why do you use the midpoint as an estimation of sea surface distance? Could you please explain more?

**AC A-3:** To estimate the distance from the scanner head to the water surface, we use the assumption that the pulse of the laser is absorbed by the water and therefore only the background noise is measured once it has passed the sea surface. The centre of the probe volume can be considered as fictitious measurement point . We also assume that the pulse intensity is symmetrical around the mid point of the probe volume and that consequently the water surface is assumed at a distance where half of the probe volume is already absorbed by the water and the signal is therefore already weaker. The midpoint of the inverted Sigmoid function indicates exactly the point between the "full" signal and a signal strength at the level of the background noise. It is not necessarily the case that when half of the probe volume is absorbed in the logarithmic representation, the signal strength also falls to exactly half in relation to the background noise, but we have been able to obtain good and consistent results with this assumption.

We added the following sentence for clarification after the definition of $r_{sea}$:

> For the estimation of the distance to the sea surface we have chosen this value, because at this distance the signal is partially weakened, which represents a partial absorption of the probe volume. This assumption is considered reasonable since the height of the lidar device above the still water level calculated by trigonometrical relations corresponds well with the actual height above the sea surface.

**RC A-4:** Line 346: What does the text "in very similar values for the angles" mean? Do you want to say that the result of the consecutive measurements has small variations with the time series? Please explain better.

**AC A-4:** The sentence would have been easier to understand if we had additionally written the word "respective" before the word "angles". We wanted to express with the sentence that the results for the roll and pitch angles are each coherent, even if there are larger fluctuations in the results over the entire series of measurements, the difference between successive measurements is very small, which suggests that the measurement uncertainty is quite small and the fluctuations over the course of the entire time series have other deterministic reasons. We change the sentence in the manuscript as follows:

> Over the entire series of measurements, larger fluctuations can be observed for both the roll angle and the pitch angle. For directly consecutive measurements, however, the changes are very small in both cases. From this we conclude that the measurement noise is low and we attribute the changes over the entire series of measurements to the variable thrust loading of the turbine at different wind speeds.

**RC A-5:** It would be good to give a suggestion on how often should we use these methods during a lidar measurement campaign. Every 10 minutes, every day??

**AC A-5:** That's a very good question, which we find difficult to answer because it can depend very much on external factors. The northing of the lidar, when installed on a fixed platform, should not actually change during a measurement campaign, so a Hard Targeting at the beginning of the measurement campaign should be sufficient. Nevertheless, it is useful and not a big effort to check the alignment with a HT scan from time to time, especially after maintenance work on or close to the device. Since unforeseen events could lead to a rotation of the device.

The same applies to SSL when the unit is installed on an immovable platform. But as we have learned, even a transition piece of a WT has enough movement to affect the measurements. Therefore, in such a case, SSL should be carried out for a longer period of time (several da to determine the necessary parameters for the PTM. Afterwards, we recommend repeating the SSL scans only occasionally (e.g. once a week) to check whether the alignment and the

parameters are still valid or whether any conditions have changed. We added the following paragraph to the discussion chapter:

.....Until then, we recommend performing the HT at the beginning of a measurement campaign to correct the orientation. Afterwards, the SSL should be run continuously to determine the levelling and to obtain sufficient data for the parameterisation of the PTM. During the campaign, it is recommended to repeat the SSL from time to time to check if the parameterisation is still acceptable and the levelling of the lidar is still correct.

**2.3 Technical correction**

**RC A-6:**

- Figure 6 and figure 10: It would be better to try to make the pictures bigger so that readers can clearly see the labels of the pictures. If you want to show two or more sub-figures in one picture, it would be clearer to give them symbols like "a)" and "b)".

- Figure 9: Please give units of the variables of the x-axis and y-axis in Figure 9.

- Figures 13, 14, and 15: The "m" in the pitch angle should be changed to subscript.

- Line 104: "probe length volume" -¿ "probe volume".

- Line 252: It should be Equation 30 instead of Equation 27.

- Line 318, 474: It seems that the right parenthesis is missing.

- Some places: "Exemplary" -¿ "Example".

- It would be better to keep the size of all pictures and text in pictures as consistent as possible, basically some plots have to have larger fonts.

**AC A-6:** Thank you for pointing out these errors. We have resized the graphics to fit the sizes specified in the latex document, so the font size is also legible and consistent. We have added the labels in the appropriate places and corrected the errors mentioned. Regarding the addition of letters to the subplots: In my experience, the letters to distinguish the sub-plots are added during the layout process when the layout is changed to a two-column layout if the paper is to be published.

**3 Referee B**

**3.1 General Comments**

**RC B-1:** The manuscript presents a couple of methods to estimate the location and orientation of a scanning lidar without extra sensors. This is an

useful study, and the results are promising. However, before I recommend the manuscript for publication, several comments need to be addressed/clarified.

**AC B1:** We thank the referee very much for the very constructive comments on our study.

**RC B-2:** The averaged height of a lidar with respect to the sea surface can be achieved using long-term data. But I am wondering about the estimated position and orientation of the lidar. Depending on the time scale of the wave motions, wave motions can significantly influence the location, tilt, and pitch every couple of seconds. Lidar measurements can take a couple of minutes to complete a targeted scan depending on the scan patterns. I am not sure what time-scale is targeted in this study while using the lidar scan data in different methods. Any explanation/clarification on this will be valuable. I was expecting a better motivation that the location, tilt, pitch information can be used to correct the lidar scan data. Is it possible to use the estimated position, pitch, and roll data from this study in the retrieval process of lidar scan data?

**AC B-2:** In order to answer the questions in this comment, it is necessary to distinguish between the different methods used in this study, in particular the two different types of scans. On the one hand, we performed "Hard Target Scans" (PPI scan with 0° elevation) to determine the position and north orientation of the lidar using the known locations of the surrounding wind turbines, and on the other hand, we performed "Sea Surface Levelling Scans" (PPI scan with -3° elevation) to measure the height above the water surface and the levelling (roll and pitch angle) of the lidar. For the hard target scans, we first performed a flat 360° PPI scan for general orientation. This allowed us to identify which hard targets are visible from the lidar at all and for which azimuth angles we detect them. We then targeted the hard targets in several separate scans, selecting the very high spatial resolution and choosing the azimuth angles and range-gates of the lidar to cover the hard target with a bit of space next to and in front of the hard target. We assume that the dynamic tilt of the transition piece has only a very small influence on the detection of the hard targets, since a slight tilt of the lidar mainly changes the height of the measurement points and less the lateral position or the distance to the hard targets. Therefore, we did not consider a temporal component for these scans. With the result of the hard targeting, the north orientation and the position of the lidar could be determined. This allows the relative position data from the lidar to be transformed into the global coordinate system.

For the Sea Surface Leveling scans, the temporal resolution of the scans is more important. With the scan configuration used in our study, the long range scanning lidar system required almost 5 minutes for a complete sea surface levelling scan. It is possible to increase the temporal resolution, but this decreases the spatial resolution, so this is a trade-off. From our SSL measurement data, the SSL method can only determine an "average" levelling for this

period. The dominant vibration period of a typical offshore support structure is in the order of a few seconds, which corresponds to the fundamental bending eigenfrequency of the structure. Components with higher frequencies are also present, but have much smaller amplitudes. Faster fluctuations/vibrations in the tilt are averaged out and not recorded. However, we assume that the influence of waves and currents on the tilt is usually much smaller than the influence of the thrust of the wind turbine. But here, too, higher-frequency fluctuations occur that cannot be measured with this method. The Platform Tilt Model, the parameterisation of which we determined with the help of Sea Surface Levelling, can theoretically also model tilt with a higher time resolution, since the tilt is determined directly from the quotient of the power and the wind speed measurements, which in our case were available in a 1 Hz resolution. For the parameterisation, however, the 5 min mean values were used. The model does not represent inertias, which would dynamically filter a direct translation and presumably damp higher-frequency vibrations, which is why we assume that the model would overestimate the fluctuations without corresponding averaging of the input values. The targeted time scale is therefore a couple of minutes. This is the typical duration for long-range scans, for which the determined levelling of the lidar is also the most interesting, since an inclination of the lidar here gives the greatest error in the height of the measured values. With the Platform Tilt Model, the levelling of the lidar can be estimated from the operating data of the wind turbine and this information can be used to correct the position of the measuring points of a scan (especially the measurement height). The temporal resolution is not sufficient to correct each individual beam of the lidar individually, but only the entire scan. To address the questions in this comment in the manuscript we added the following sentence to the introduction:

> The only drawback of this method is that the levelling is based on the high resolution PPI scan, which takes a few minutes with conventional systems. Therefore, higher frequency fluctuations are not captured with this method, but only an average estimate in the few-minute timescale can be produced.

We added the following sentence to the conclusions of the paper:

> This information can be used to correct the position data of the measuring points of ongoing lidar measurements. A more precise orientation of the lidar helps to transfer the measurement points more accurately into the global coordinate system, and a more precise levelling provides a better estimate of the actual height at which the wind speed was measured with a lidar. This helps to reduce uncertainties in long range scanning lidar data analysis.

**RC B-3:** Depending on the pulse shape and range gate length, the detected location of the hard target will vary. Is the center of range gate along the line-of-sight moved to find the accurate location of the hard target? It is not possible to obtain an accurate position of a hard target based on a single range gate around the hard target (example: check Figure 2, Choukulkar et al. 2017). The post-processing of the lidar data to obtain the exact location of the hard target is not clear.

**AC B-3:** Along the line of sight we measured at several distance measuring points with high resolution. These points represent the centre of the probe volume, so that the probe volumes of the different measurement locations overlap. A solid object therefore shows up in the form of a high cnr value at several adjacent measurement points and it is not possible to determine the exact position of a hard target. In addition, the hard targets used in our case (the tower of the surrounding wind turbines) have a diameter that we do not know exactly. It is possible that our scan only hits the tower or also the much wider platform on the transition piece. For this reason, we have represented the hard targets, i.e. the physical object for which we know the coordinates, not only by a single hard target measuring point, but by a cluster of measuring points, i. e. all measuring points that fulfil the criterion from Equation (1). Depending on the distance to the object and the size of the object, these are several measuring points in line-of-sight behind each other and possibly the object is even detected on several neighbouring beams. For example, in Figure 8 the measuring points for which the criterion is fulfilled at the northernmost detected turbine are shown as orange dots in the zoomed-in window. We do not know which measuring point best corresponds to the centre of the hard target, but we do not need to know this because the algorithm presented tries to minimise the distances of all hard target measuring points to the known coordinates. In order to better explain how the algorithm works, we have changed the sentence in line 114f to read as follows:

> If the resolution of the scan is high enough, a solid object, such as the tower of a wind turbine, is represented by a cluster of measurement points, which meet the criterion from Equation (1).

**RC B-4:** Depending on the accuracy of the angular resolution of the lidar, the location (and other estimated variables) of a hard target will vary. For example, there could be a backslash problem with the lidar. Maybe the pointing accuracy of 0.1 deg is not maintained for a long field campaign. It is recommended to do scans in raster mode (same scan with increasing azimuth and decreasing azimuth) to check the accuracy of the angular scan. It would be valuable to discuss the impact of angular resolution on the results of this study.

**AC B-4:** That is a very interesting point. In this study, however, we did not want to focus too much on device-specific malfunction. For the hard target scans, we used the highest resolution for the azimut angle in order to be able to determine the alignment as accurately as possible. To minimise possible errors in the azimuth angle during faster scans, it is advisable to quantify them with the methods you mentioned. [Vasiljević, N. (2014). A time-space synchronization of coherent Doppler scanning lidars for 3D measurements of wind fields (Vol. 0027)] gives further guidance on increasing the quality of measurements and discusses the backlash problem and how to . For the Sea Surface Leveling scans, a small offset error in the azimuth angle of the lidar would result in a slight rotation of the intersection of the scan with the water surface. For small

rotations, the values derived from this for the roll and pitch angles change only minimally, so the error was not considered further. To adress this point in the manuscript, we added the following paragraph to the discussion:

> In connection with hard target scans, it is recommended to investigate the pointing accuracy of the lidar's scanner unit as disscussed in [Vasiljević, N. (2014). A time-space synchronization of coherent Doppler scanning lidars for 3D measurements of wind fields (Vol. 0027).]. Depending on the device or device type, there may be different sources of inaccuracies, such as the backlash problem. It is therefore advisable to quantify such errors before a hard target scan by performing scans with different scan speeds or scan rotation directions.

**RC B-5:** There are a couple of typos in the manuscript. I suggest checking the article carefully before the next submission. Also, remove the italic font such as "Platform Tilt Model".

**AC B-5:** Thank you very much for your comment. The manuscript will be checked again for typing errors.

**3.2 Specific Comments**

**RC B-6:** L39: Not clear with this sentence "Doppler wind lidars are able to detect even very little backscatter from aerosols."

**AC B-6:** We point out that the sensor unit of a Doppler wind lidar can measure the backscatter of the laser light sent back by aerosols. In contrast, the backscatter from solid objects is many times higher and is characterised in the signal by a very high CNR value. To better express this point, we change the wording of this sentence to:

> Doppler wind lidars are able to detect the backscatter of the laser beam from aerosols. Therefore, measurements against.....

**RC B-7:** L114 and L115: explain "a cluster of hard target measurements"

**AC B-7:** As mentioned above, the set $L_{\mathrm{ht}}$ is defined in such a way that for a larger hard target, several adjacent measuring points are classified as a hard target measuring point. This means that an object is represented as a cluster of measuring points. As already mentioned above we changed the corresponding sentence as follows:

> If the resolution of the scan is fine enough, a solid object, such as the tower of a wind turbine, is represented by cluster of measurement points, which meet the criterion from Equation (1).